# AdaProb: Towards Efficient Machine Unlearning via Adaptive Probability

## Abstract

Machine unlearning—enabling a trained model to forget specific data—is crucial for addressing erroneous data and adhering to privacy regulations like the General Data Protection Regulation (GDPR)'s "right to be forgotten." Despite recent progress, existing methods face two key challenges: residual information may persist in the model even after unlearning, and the computational overhead required for effective data removal is often high. To address these issues, we propose Adaptive Probability Approximate Unlearning (AdaProb), a novel method that enables models to forget data efficiently and in a privacy-preserving manner. Our method firstly replaces the neural network's final-layer output probabilities with pseudo-probabilities for data to be forgotten. These pseudo-probabilities follow a uniform distribution to maximize unlearning, and they are optimized to align with the model's overall distribution to enhance privacy and reduce the risk of membership inference attacks. Then, the model's weights are updated accordingly. Through comprehensive experiments, our method outperforms state-of-the-art approaches with over 20% improvement in forgetting error, better protection against membership inference attacks, and less than half the computational time.[1]

## 1 Introduction

Machine unlearning focuses on eliminating the impact of specific data subsets—such as erroneous, or privacy-leaking instances (Jagielski et al., 2018; Yang et al., 2024)—used in model training (Baumhauer et al., 2022; Fu et al., 2022; Golatkar et al., 2020a;b; Guo et al., 2019; Kim & Woo, 2022; Mehta et al., 2022; Nguyen et al., 2020; Shah et al., 2023). It has emerged as a critical area of research due to growing concerns about data privacy (Pardau, 2018), legal requirements for data deletion (Mantelero, 2013), and the need for models to adapt to new information without complete retraining. Though the most straightforward approach is to retrain the model with a new dataset that excludes the data needing removal, this approach is computationally expensive and needs continuous access to the entire training set.

Two of the most prominent use cases for machine unlearning are privacy protection and removal of poisoned, biased, or erroneous data (Nguyen et al., 2022). In privacy protection, unlearning aims to modify the model to forget a set of training data points, so that an adversary cannot determine anything about them from the model, including whether or not they were part of the training set (Hu et al., 2024). Conventional unlearning methods often fail to achieve this behavior: in forcing the model to perform poorly on the forget set (i.e., exhibit high loss), they create a distinguishable pattern between forgotten and retained data (Wang et al., 2024; Chen et al., 2021). This performance disparity enables attackers to identify forgotten samples through techniques like Membership Inference Attacks (MIA) (Hui et al., 2021). To prevent this vulnerability, effective privacy-preserving unlearning must ensure the model's behavior on forgotten data is indistinguishable from its behavior had it never encountered that data during training (Guo et al., 2019; Xu et al., 2023). This often requires sacrificing some model performance to achieve stronger privacy protection (Qu et al., 2023). The second objective, poisoned data removal, differs fundamentally from privacy protection. Machine learning models are vulnerable to accidentally erroneous or intentionally poisoned training data, where malicious data is injected into the training set, causing the model to produce unsafe or incorrect outputs (Mehrabi et al., 2021). Unlike privacy-focused unlearning, which aims to protect

---

[1]The code is provided in https://anonymous.4open.science/r/AdaProb-BC78/

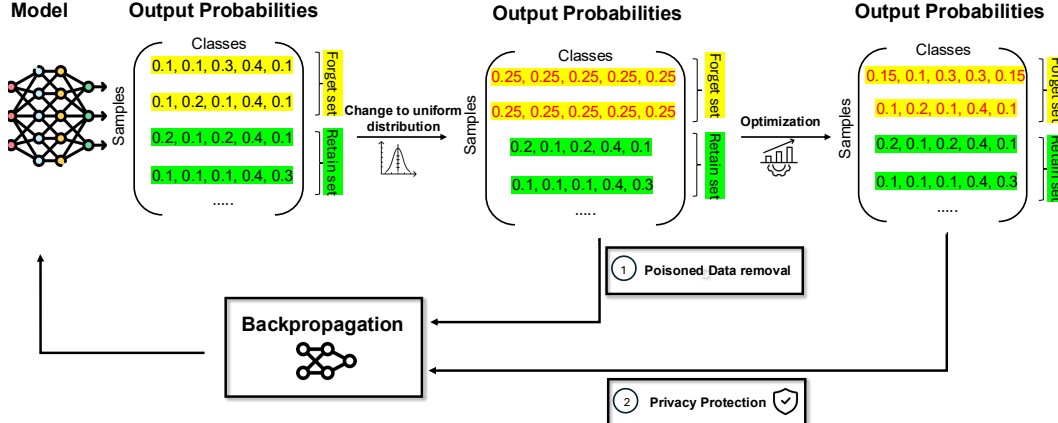

Figure 1: This is an overview of Adaptive Probability Unlearning (AdaProb). In this approach, we extract the output layer probabilities and replace the forget set probabilities with pseudo-probabilities. After performing optimization, the model's weights are fine-tuned using the refined pseudo-probabilities.

individual data points from being inferred, poisoned data removal focuses on eliminating corrupted samples while maintaining or even improving the model's overall performance. In this context, the goal is to minimize the model's accuracy on the forget set (containing poisoned data) while preserving high accuracy on clean data. Notably, vulnerability to membership inference attacks (MIAs) is not a concern, as the priority is correcting model behavior.

Current machine unlearning methods primarily focus on either gradient-based approaches (Neel et al., 2021) that optimize modified loss functions to induce forgetting or adjusting the model architecture through layer addition/deletion. We propose a fundamentally different approach that manipulates the final-layer output probabilities and leverages backpropagation to update model weights accordingly.

Our Adaptive Probability Approximate Unlearning (AdaProb) method replaces the model's output probabilities with uniformly distributed pseudo-probabilities for forget-set data, ensuring effective forgetting. The method operates differently in two settings: In poisoned data removal, these pseudo-probabilities are backpropagated to update model weights. In privacy protection, we iteratively refine the output probability distribution to align forget-set outputs with pseudo-probabilities while constraining retain-set outputs to remain similar to the original model's predictions. This dual constraint ensures forgotten samples become indistinguishable from retained data, preventing information leakage. The overview of the method is illustrated in Figure 1.

Our extensive evaluations demonstrate that AdaProb achieves a 50% reduction in computational time compared to state-of-the-art methods while simultaneously improving unlearning effectiveness. Notably, AdaProb reduces the success rate of membership inference attacks to near-random guessing levels, validating its strong privacy protection capabilities.

Our contributions are threefold: (1) We propose a novel unlearning method based on output probability manipulation that handles both privacy and poisoning data removal tasks. (2) We achieve 50% computational speedup over existing methods while improving unlearning performance (3) We provide comprehensive experimental validation demonstrating superior privacy protection, with membership inference attacks reduced to random-guessing levels. Additionally, our method achieves over 20% improvement in forget set error for poisoned data removal tasks.

## 2 RELATED WORK

**Machine Unlearning** Machine unlearning, first proposed by Cao & Yang (2015), has evolved into two main paradigms: exact unlearning, which ensures complete data removal, and approximate unlearning, which reduces data influence to acceptable levels (Izzo et al., 2021). While exact un-

learning methods have been developed for specific models (Brophy & Lowd, 2021; Schelter et al., 2021; Ginart et al., 2019), they suffer from prohibitive computational costs, especially as model size increases.

Approximate unlearning methods have been developed to address the computational challenges of high-dimensional neural networks. These approaches employ various strategies: weight modification methods directly adjust model parameters (Golatkar et al., 2020a;b), architectural approaches like SISA training partition data during training to facilitate removal (Bourtoule et al., 2021), while others leverage cached gradients (Wu et al., 2020) or optimization techniques (Kurmanji et al., 2024) to accelerate retraining. Additionally, certified unlearning methods formalize the unlearning goal by requiring the unlearned model to be provably close to one retrained from scratch on only the retained data (Zhang et al., 2024). However, these approaches face critical limitations: they are still computationally expensive, often compromise model utility, and—most importantly—fail to address privacy protection against membership inference attacks perfectly.

This gap motivates our approach: rather than modifying parameters directly or restructuring training, we manipulate output probabilities to achieve efficient unlearning while providing strong privacy guarantees. Our method addresses the key shortcomings of existing work by reducing computational time by 50% and explicitly protecting against privacy attacks, achieving near-random membership inference success rates.

**Machine Unlearning Evaluations**   To evaluate unlearning methods, it is common to compare models before and after unlearning across three key metrics: computational efficiency, model utility, and privacy protection. Computational efficiency is measured by the time required for the unlearning process, while model utility is assessed by comparing test set performance before and after unlearning. Privacy protection, however, is more challenging to measure. Current approaches include: (1) comparing posterior distributions or parameters between retrained and unlearned models (Golatkar et al., 2020a;b), (2) providing theoretical guarantees or bounds for the unlearned model (Chien et al., 2022; Guo et al., 2019; Neel et al., 2021), and (3) applying attacks to measure privacy risks (Chen et al., 2021), such as membership inference attacks (MIA) (Shokri et al., 2017), which use shadow models to generate synthetic training data for attack classifiers. In the paper, we measure the privacy protection through membership inference attack, and the KL divergence of output distribution on the forget set between the retrained model and the unlearned model.

## 3 NOTATIONS AND PROBLEM DEFINITION

Consider a dataset $(\mathbf{x}, y) \in \mathcal{D}$, composed of $N$ data points, where each instance consists of an input feature vector $\mathbf{x}_i$ and its corresponding label $y_i$. Let $f(\cdot; \mathbf{w})$ represent a function implemented by a deep neural network, parameterized by the weights $\mathbf{w}$. In this context, we are provided with a "forget set" $\mathcal{D}_{\mathrm{f}} = (\mathbf{x}_i, y_i)_{i=1}^{N_f} \subset \mathcal{D}$, consisting of $N_f$ instances extracted from $\mathcal{D}$, as well as a "retain set" $\mathcal{D}_{\mathrm{r}} = (\mathbf{x}_j, y_j)_{j=1}^{N_r} \subset \mathcal{D}$ containing $N_r$ training samples. For simplicity, we assume that $\mathcal{D}_{\mathrm{r}}$ is the complement of $\mathcal{D}_{\mathrm{f}}$, satisfying the condition $\mathcal{D}_{\mathrm{f}} \cup \mathcal{D}_{\mathrm{r}} = \mathcal{D}$ and $N_f + N_r = N$, thereby covering the entire original dataset.

The goal of machine unlearning is to derive a new set of weights, $\mathbf{w}_u$, such that the updated model, $f(\cdot; \mathbf{w}_u)$, effectively forget the information related to $\mathcal{D}_{\mathrm{f}}$. Specifically, the unlearned model should maintain its original performance on the retain set $D_{\mathrm{r}}$ and retain its ability to generalize to unseen data. In the paper, we use "original" to indicate the pre-unlearning model.

## 4 METHODS

Building on the foundational framework, we propose a machine unlearning approach that optimizes the output probabilities at the final layer and subsequently backpropagates these adjustments to update the model weights throughout the network.

We define the output layer probability distribution for each data point as a $k$-dimensional vector, where $k$ is the number of classes. Let $f(x, \mathbf{w})$ denote the output probability distribution generated

when input $x$ is passed through the model with weights $\mathbf{w}$. For the forget set $D_{\mathrm{f}}$ and retain set $D_{\mathrm{r}}$, we denote their output distributions as $\{f(x_i; \mathbf{w})\}_{i=1}^{N_f}$ and $\{f(x_j; \mathbf{w})\}_{j=1}^{N_r}$, respectively.

The core of our method lies in formulating an optimization objective that adjusts the model's output distribution to effectively forget the information in the forget set $\mathcal{D}_{\mathrm{f}}$ while preserving performance on the retain set $\mathcal{D}_{\mathrm{r}}$. We first change the forget set output distributions to uniform distributions to obscure learned patterns, and keep the original model output distribution for the retain set to maintain performance. Then, the optimization minimizes the discrepancy between current outputs $g(x; \mathbf{w})$ and original distributions $f(x; \mathbf{w})$. After obtaining the optimal output distributions, we backpropagate to update the model weights, teaching the network to realize these adjusted predictions.

### 4.1 Pseudo-Probability Refinement

To find the optimal output distributions, we start by replacing the model's output distribution with a pseudo-probabilistic distribution, such as a uniform distribution. The rationale behind this strategy is to "mask" or obscure the model's learned associations with the forget set by assigning equal probabilities to each class, thereby eliminating the model's ability to make confident predictions on these data points.

Specifically, we construct a probability matrix, where each row represents an input data point and each column represents a class. For a dataset with $N$ data points and $K$ classes, the matrix has dimension $N \times K$. Each element $(i, k)$ contains $g_k(x_i; \mathbf{w})$, the probability of data point $x_i$ belonging to class $k$. The matrix representation facilitates our optimization constraints on both row sums and column sums.

In our formulation, $\{g(x_i, \mathbf{w})\}_{i=1}^{N_f}$ denotes the uniform pseudo-probability distribution for the forget set $D_{\mathrm{f}}$. These are designed to disrupt the model's learned patterns while preserving performance on the retain set. For the retain set, $\{g(x_j, \mathbf{w})\}_{j=1}^{N_r}$ is set to the original model outputs $\{f(x_j, \mathbf{w}_{\mathrm{original}})\}_{j=1}^{N_r}$. Given a data point $x_i$ in the forget set, $g_k(x_i, \mathbf{w})$ denotes its probability of belonging to class $k$, where $k \in \{1, ..., K\}$

In the data poisoning setting, after changing the forget set output distribution to uniform probabilities, the model weights are directly updated. However, in the privacy setting, directly using uniform distributions makes the model vulnerable to membership inference attacks, as such artificial patterns are easily detectable. We apply our optimization to minimize the KL divergence between current and original distributions for both sets, balanced by parameter $\lambda$. This can let the forget set output distribution to be similar to the original distribution, which makes it hard to be detected by membership inference attack.

To address this, we introduce constraints on our probability matrix: (1) **Column constraints:** The sum of each column (total probability for class $k$ across all data points) must equal $M_k = \sum_{i=1}^{N_f} g_k(x_i; \mathbf{w}) + \sum_{j=1}^{N_r} g_k(x_j; \mathbf{w})$ (2) **Row constraints:** Each row must sum to 1, ensuring valid probability distributions. (3) **Element constraints:** All probabilities must lie in $[0, 1]$.

The optimization updates the output distribution $\{g(x_i, \mathbf{w})\}_{i=1}^{N_f}$ for the forget set and $\{g(x_j, \mathbf{w})\}_{j=1}^{N_r}$ for the retain set to find the optimal values that minimize the objective function while satisfying all constraints.

$$\min_{\{g(x_i; \mathbf{w})\}_{i=1}^{N_f}, \{g(x_j; \mathbf{w})\}_{j=1}^{N_r}} \left( \sum_{i=1}^{N_f} D_{KL}(g(x_i, \mathbf{w}) \| f(x_i; \mathbf{w})) + \lambda \sum_{j=1}^{N_r} D_{KL}(g(x_i, \mathbf{w}) \| f(x_j; \mathbf{w})) \right) \quad (1)$$

$$\text{subject to} \quad \sum_{i=1}^{N_f} g_k(x_i; \mathbf{w}) + \sum_{j=1}^{N_r} g_k(x_j; \mathbf{w}) = M_k, \quad \forall k \in \{1, ..., K\}, \quad (2)$$

$$\sum_{k=1}^{K} g_k(x_i; \mathbf{w}) = 1, \forall i \in \{1, ..., N_f\}, \quad \sum_{k=1}^{K} g_k(x_j; \mathbf{w}) = 1, \forall j \in \{1, ..., N_r\} \quad (3)$$

$$g_k(x_i; \mathbf{w}) \in [0, 1], \forall i \in \{1, ..., N_f\}, \forall k, \quad g_k(x_j; \mathbf{w}) \in [0, 1], \forall j \in \{1, ..., N_r\}, \forall k \quad (4)$$

### 4.1.1 CONVERGENCE TO THE UNIQUE OPTIMAL SOLUTION

To address computational efficiency for large datasets, we develop an iterative algorithm based on coordinate descent applied to our constrained optimization problem.

**Theorem 1.** *The proposed iterative procedure converges to the unique optimal solution, provided that feasible initial conditions are used and the KL divergence remains finite for all feasible distributions.*

*Proof sketch:* The KL divergence $D_{\mathrm{KL}}(p\|q)$ is strictly convex in $p$ when $q$ is fixed. Since our objective function is a sum of strictly convex functions, it is strictly convex overall. Combined with linear constraints, this yields a convex optimization problem with a unique global minimum. Our coordinate descent method maintains feasibility through closed-form updates, guaranteeing convergence to the global optimum.

## 4.2 OPTIMIZATION ALGORITHM

We solve the constrained optimization problem using coordinate descent with Lagrangian multipliers.

### 4.2.1 LAGRANGIAN FORMULATION

To handle the constraints, we introduce Lagrange multipliers $\alpha_k$ for each class $k$:

$$
\begin{aligned}
\mathcal{L}(f, \alpha) = {} & \sum_{i=1}^{N_f} D_{\mathrm{KL}}\big(g(x_i; \mathbf{w}) \,\|\, f(x_i; \mathbf{w})\big) + \lambda \sum_{j=1}^{N_r} D_{\mathrm{KL}}\big(g(x_j; \mathbf{w}) \,\|\, f(x_j; \mathbf{w})\big) \\
& + \sum_{k=1}^{K} \alpha_k \left( \sum_{i=1}^{N_f} g_k(x_i; \mathbf{w}) + \sum_{j=1}^{N_r} g_k(x_j; \mathbf{w}) - M_k \right)
\end{aligned}
\tag{5}
$$

### 4.2.2 COORDINATE DESCENT UPDATES

Taking derivatives and applying KKT conditions yields the closed-form updates:

**Primal updates:**

$$
g_k(x_i; \mathbf{w}) = \hat{g}_{i,k} \exp(-\alpha_k), \quad g_k(x_j; \mathbf{w}) = \hat{g}_{j,k} \exp(-\alpha_k/\lambda)
\tag{6}
$$

**Dual updates:**

$$
\alpha_k^{(t+1)} = \alpha_k^{(t)} + \eta \left( \sum_{i=1}^{N_f} g_k^{(t)}(x_i; \mathbf{w}) + \sum_{j=1}^{N_r} g_k^{(t)}(x_j; \mathbf{w}) - M_k \right)
\tag{7}
$$

where $\eta > 0$ is the step size. The algorithm alternates between these updates until convergence.

## 4.3 WEIGHT UPDATE VIA BACKPROPAGATION

After obtaining optimal output distributions through the above optimization, we update the model weights to realize these target distributions. We define a loss function based on the KL divergence:

$$
\mathcal{L}_{\mathrm{weight}} = \sum_{i=1}^{N_f} D_{\mathrm{KL}}(g^*(x_i)\|f(x_i; \mathbf{w})) + \sum_{j=1}^{N_r} D_{\mathrm{KL}}(g^*(x_j)\|f(x_j; \mathbf{w}))
\tag{8}
$$

where $g^*$ denotes the optimal distributions from our optimization. The weights are updated via gradient descent:

$$\mathbf{w}^{(t+1)} = \mathbf{w}^{(t)} - \gamma \nabla_{\mathbf{w}} \mathcal{L}_{\text{weight}} \tag{9}$$

This ensures the model's outputs converge to the optimized distributions that achieve unlearning while maintaining natural probability patterns.

## 5 EXPERIMENT

### 5.1 DATASETS AND METRICS

In this study, we employ three datasets that were also used in prior research: CIFAR-10, CIFAR-100, and Lacuna-10. Lacuna-10 is a curated dataset formed by selecting data from 10 distinct classes, randomly chosen from the extensive VGG-Face2 dataset (Cao et al., 2018). These selected classes each have a minimum of 500 samples, with the data further segmented into 400 training and 100 testing images per class. Lacuna-100 expands on this concept by selecting 100 classes with the same criteria.

Our evaluation employs multiple metrics to comprehensively assess unlearning performance. We measure the model's error rate (defined as $100\% -$ accuracy) on three sets: the forget set to verify successful unlearning, the retain set to evaluate memory preservation, and the test set to assess generalization ability. For privacy protection tasks, we additionally evaluate the model's resistance to membership inference attacks. We also introduce a metric that measures the KL divergence between the output distributions of the unlearned and retrained models on forget set inputs, evaluating how closely the unlearned model approximates ideal retraining behavior.

### 5.2 IMPLEMENTATION DETAILS

To facilitate comprehensive comparison with the performance of other models, we follow the setup in (Kurmanji et al., 2024). We establish two experimental conditions: small-scale and large-scale. The small-scale setting, referred to as CIFAR-5/Lacuna-5, involves a subset of 5 classes from each dataset, comprising 100 training, 25 validation, and 100 testing samples per class. Notably, the forget set includes 25 samples from the initial class, accounting for 5% of the dataset. Conversely, the large-scale setting encompasses all classes from both CIFAR-10 and Lacuna-10, providing a broader spectrum for analysis. In the large-scale scenario, we will explore both class unlearning and selective unlearning. For class unlearning, we define the forget set as the entirety of the training set for class 5, which constitutes 10% of the data. In the selective unlearning scenario, we aim to forget 100 examples from class 5, representing 0.25% of CIFAR-10 and 2% of Lacuna-10.

To align with precedents in the field, our experiments are conducted using two architectures: ResNet-18 and ALL-CNN (Springenberg et al., 2014). The baseline model is pretrained on the CIFAR-100 and Lacuna-100 datasets for initial weight setting. Additionally, $\lambda$ is be set to a default value of 1 in the following experiments. More details of hyperparameters are are shown in Appendix E.

### 5.3 BASELINE

Our approach is benchmarked against the latest state-of-the-art methods and established baselines to highlight its efficacy: **Retrain**: The model is trained solely on the retain set $\mathcal{D}_r$, considered the gold standard. However, this method is typically deemed impractical for real-world applications. **Original**: The baseline model is trained on the complete dataset $\mathcal{D}$, without any modifications for data forgetting. **Finetuning**: The original model is fine-tuned on the retain set $\mathcal{D}_r$, incorporating no specific forgetting mechanism. **NegGrad+** (Kodge et al., 2023): An innovative method that applies gradient ascent to the forget set and gradient descent to the retain set over 500 iterations. **Fisher Forgetting** (Golatkar et al., 2020a): Adjusts the model's weights to effectively "unlearn" the data meant to be forgotten, simulating a scenario where the model was never exposed to this data. **NTK Forgetting** (Doan et al., 2021): Employs novel techniques like PCA-OGD to minimize forgetting by orthogonally projecting onto principal directions, preserving data structure integrity. **CF-k, EU-k** (Goel et al., 2022): These methods focus on the model's last k layers. "Exact-unlearning" (EU-k) re-trains these layers from scratch, while "Catastrophic Forgetting" (CF-k) fine-tunes them on

Table 1: Unlearning results with ResNet-18 for the poisoning data removal task. Our method achieves higher forget rates while preserving overall model performance. We evaluate two variants: "AdaProb w/ uniform" uses uniformly distributed pseudo-probabilities, while "AdaProb w/ random" employs randomly sampled pseudo-probabilities. All values are reported as percentages, where each number represents the error rate (100% - accuracy).

| Model | CIFAR-5 | | | Lacuna-5 | | |
|---|---|---|---|---|---|---|
| | Test error | Retain error ($\downarrow$) | Forget error ($\uparrow$) | Test error | Retain error ($\downarrow$) | Forget error ($\uparrow$) |
| Retrain | 24.90 | **0.00** | 28.80 | 5.80 | **0.00** | 4.80 |
| Original | 24.20 | **0.00** | 0.00 | 5.70 | **0.00** | 0.00 |
| Finetune | 24.30 | **0.00** | 0.00 | 5.60 | **0.00** | 0.00 |
| Fisher | 31.60 | 14.00 | 4.80 | 6.70 | 14.00 | 6.40 |
| NTK | 24.40 | **0.00** | 22.40 | 5.60 | **0.00** | 0.00 |
| NegGrad+ | 25.50 | **0.00** | 41.3 | 6.10 | **0.00** | 1.30 |
| CF-k | 22.60 | **0.00** | 0.00 | 5.80 | **0.00** | 0.00 |
| EU-k | 23.50 | **0.00** | 10.70 | 5.90 | **0.00** | 0.00 |
| Bad-T | 22.73 | 5.12 | 8.00 | 5.00 | 8.64 | 0.14 |
| SCRUB | 24.20 | **0.00** | 40.80 | 6.20 | **0.00** | 24.80 |
| **AdaProb w/ random** | **22.00** | 0.00 | **80.00** | **2.20** | 0.00 | 64.00 |
| **AdaProb w/ uniform** | 27.00 | 0.21 | **80.00** | 2.80 | 0.42 | **68.00** |

Table 2: KL divergence between output distributions of unlearned and retrained models on forget set inputs. A lower KL divergence indicates closer alignment with the retrained model's output distribution, providing better privacy protection.

| Task | KL(AdaProb$\|$Retrain) ($\downarrow$) | KL(SCRUB$\|$Retrain) ($\downarrow$) |
|---|---|---|
| ResNet on Lacuna-5 | 1.35 | 3.65 |
| ResNet on Lacuna-10 | 5.76 | 5.88 |
| ResNet on CIFAR-5 | 2.56 | 3.01 |
| ResNet on CIFAR-10 | 7.89 | 7.79 |
| ALLCNN on Lacuna-5 | 2.02 | 2.23 |

the retain set $\mathcal{D}_r$. **SCRUB** (Kurmanji et al., 2024): Introduces a novel training objective and has demonstrated superior performance in prior metrics.

## 5.4 POISONED DATA REMOVAL

For poisoned data removal, our objective is to maximize the forget set error rate. Rather than performing complex optimizations, we directly assign pseudo-probabilities to the forget set samples and update model weights through backpropagation. We investigate two pseudo-probability distributions: uniform distribution, where all classes receive equal probability, and random distribution, generated by applying the softmax function to randomly sampled values. This straightforward approach effectively corrupts the model's predictions on poisoned data while maintaining computational efficiency.

As shown in Table 1, our method AdaProb achieves a much higher forget error than other approaches (60-80 percent higher), which is the desired outcome in the poison data removal task. The uniform distribution performs better than the random distribution in terms of forget error, but it leads to worse results in test error and retain error. Table 5 demonstrates that AdaProb also achieves better performance in forget error for selective unlearning in larger models. Based on the forget error metric, our method appears to be the most successful at unlearning, achieving the desired outcome of complete forgetfulness without severely compromising the performance of the retain set. Additionally, our method exhibits the lowest test error, demonstrating that the model's performance and generalizability are well-preserved even after applying our unlearning technique. The results demonstrate that using pseudo-probabilities is effective for poisoned data removal. More poison data removal task results are shown in the Appendix A.1.

## 5.5 PRIVACY PROTECTION

In the privacy protection setting, our goal is to ensure that the forget error remains close to that of retraining to avoid leakage of privacy. We evaluate privacy protection through membership inference attacks, which is adopted from the approach outlined by Kurmanji et al. (2024). Specifically,

Table 3: Membership inference attack results with ResNet-18 and ALL-CNN in large-scale unlearning. The closer the result is to 50%, the better the performance.

| | ResNet | | | | ALL-CNN | | | |
| | Class | | Selective | | Class | | Selective | |
| Model | mean | std | mean | std | mean | std | mean | std |
| --- | --- | --- | --- | --- | --- | --- | --- | --- |
| Retrain | 49.33 | 1.67 | 54.00 | 1.63 | 55.00 | 4.00 | 48.73 | 0.24 |
| Original | 71.10 | 0.67 | 65.33 | 0.47 | 66.50 | 0.50 | 71.40 | 0.70 |
| Finetune | 75.57 | 0.69 | 64.00 | 0.82 | 68.00 | 1.00 | 74.97 | 1.27 |
| NegGrad+ | 69.57 | 1.19 | 66.67 | 1.70 | 72.00 | 0.00 | 70.03 | 1.92 |
| CF-k | 75.73 | 0.34 | 65.00 | 0.00 | 69.00 | 2.00 | 72.93 | 1.06 |
| EU-k | 54.20 | 2.27 | 53.00 | 3.27 | 66.50 | 3.50 | 51.60 | 1.22 |
| Bad-T | 54.00 | 1.10 | 59.67 | 4.19 | 63.40 | 1.20 | 77.67 | 4.11 |
| SCRUB | 52.20 | 1.71 | 78.00 | 2.45 | 52.00 | 0.00 | 54.30 | 2.24 |
| SCRUB+R | 52.20 | 1.71 | 58.67 | 1.89 | **52.00** | 0.00 | 54.30 | 2.24 |
| **AdaProb** | **51.00** | 1.05 | **58.00** | 0.93 | 54.00 | 0.70 | **50.00** | 0.40 |

we train a binary classifier (the "attacker") using the losses of the unlearned model on both the forget and test examples, with the objective of classifying instances as either "in" (forget) or "out" (test). The attacker then predicts labels for held-out losses—losses that were not used during training—balanced between the forget and test sets. A successful defense is indicated by an attacker's accuracy of 50%, signifying that the attacker is unable to distinguish between the two sets, demonstrating the effectiveness of the unlearning method.

According to Table 11, AdaProb's forget error is very close to that of retraining, particularly in the Lacuna-10 experiment, where it is the closest match. In the membership inference attack experiment, shown in Table 3, AdaProb consistently achieves nearly 50% accuracy, indicating strong privacy preservation. This demonstrates that, with the refinement of pseudo-probabilities, the model can maintain the original distribution while effectively forgetting the designated forget set. More forget error results are presented in Appendix B.

We evaluate the similarity between unlearned and retrained models by measuring the KL divergence between their output distributions on forget set inputs, using SCRUB as a baseline. As illustrated in the t-SNE visualization in Figure 2, the output probabilities of AdaProb (purple points) cluster more closely to those of the retrained model (yellow points) compared to SCRUB (blue points) in ALLCNN trained on Lacuna-5. In other settings, our method produces output distributions comparable to SCRUB. Table 2 presents KL divergence values that support this observation, showing that AdaProb achieves output distributions closer to the retrained model in certain cases, while matching SCRUB's performance in others. This demonstrates that AdaProb consistently approximates the behavior of a model that was never exposed to the forgotten data, performing at least as well as SCRUB across different scenarios. When considering the significantly reduced computation time and enhanced resistance to membership inference attacks, AdaProb emerges as the superior method. Additionally, Table 12 reports the KL divergence values between output distributions on the test set, further validating that our approach has better privacy protection. More results of KL divergence is reported in Appendix C.

## 5.6 COMPUTATIONAL EFFICIENCY

We compare the time required for SCRUB (Kurmanji et al., 2024), retraining, and our method, with all experiments conducted on an NVIDIA RTX-4090. Time is recorded over 5 runs, and we report both the mean and the standard error. In Figure 3, we present the time required for the poisoning data removal tasks using the ResNet-18 model and selective unlearning using ALL-CNN. Compared to other methods, AdaProb significantly reduces computation time, cutting it to less than half of what is required by SCRUB. The results further emphasize the high effectiveness of the optimization approach and the use of pseudo-probabilities to fine-tune the model weights.

## 5.7 ABLATION STUDIES

We conduct two further ablation studies. First, in the optimization objective function (1), the value of $\lambda$ was set to 1 in all previous experiments. In Table 4, we explore the impact of varying $\lambda$ on the retain and forget errors in a small-scale unlearning experiment on CIFAR-5 with ResNet. As

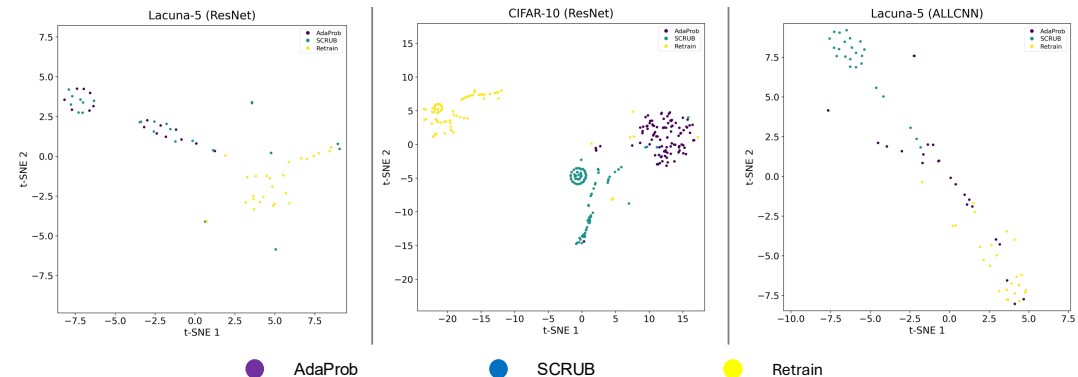

Figure 2: t-SNE map of output distributions on the forget set for Retrain, SCRUB, and our unlearned model. Points closer to the Retrain cluster indicate stronger privacy (better unlearning).

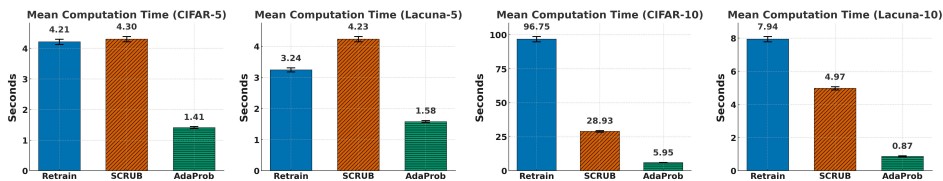

Figure 3: Time needed for the unlearning method (measured over 5 runs)

Table 4: The retain error and forget error with varying $\lambda$ values were evaluated in a small-scale unlearning experiment on CIFAR-5 using ResNet.

| Model | $\lambda = 1$ | | $\lambda = 2$ | | $\lambda = 3$ | | $\lambda = 4$ | |
|---|---|---|---|---|---|---|---|---|
| | Retain error | Forget error | Retain error | Forget error | Retain error | Forget error | Retain error | Forget error |
| **AdaProb** | 0.21 | 80.00 | 0.00 | 56.00 | 0.00 | 23.00 | 0.00 | 30.00 |

$\lambda$ increases, more weight is assigned to the retain set, resulting in a decrease in retain error from 0.21% to 0%. However, this reduction comes at a significant cost to the forget error.

Second, we investigate our method in a larger setting, using the CIFAR-100 dataset with one class unlearning. Our method demonstrated very good performance. Using the ResNet architecture, SCRUB achieved a forget error of 5.19% and a retain error of 0.00%. In contrast, our method achieved a retain error of 0.00% and a forget error of 98.25%. The details are shown in Table 13.

## 6 CONCLUSION

This research introduces a novel approach to machine unlearning, presenting an optimization framework that refines output probability distributions within deep learning models. Our method excels in striking an optimal balance between forgetting effectiveness and preserving model performance. Additionally, it demonstrates superior resilience against membership inference attacks. Empirical results across diverse datasets and model architectures, including CIFAR-10 and Lacuna-10 with ResNet and ALL-CNN, highlight the superiority of our approach over existing methods.

Furthermore, the operational flexibility, theoretical insights, and high computational efficiency of our approach provide a solid foundation for further developments. However, we acknowledge certain limitations. Our current method is limited to addressing unlearning in classification tasks and may encounter convergence issues during the optimization process. Additionally, the approach is restricted to supervised learning settings and does not extend to unsupervised tasks at this stage. Future work will focus on extending the method to various models, including large language models, and broadening its applicability beyond classification tasks.

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

## A  LLM USAGE

We used OpenAI's ChatGPT-5 to perform grammar checking on this manuscript.

### A.1  MORE RESULTS ON POISONED DATA REMOVAL

Here, we present additional results for our method on poisoned data removal tasks. In this evaluation framework, optimal performance is characterized by high error rates on the forget set (indicating successful unlearning) and low error rates on both the test and retain sets (preserving model utility). The results are incorporated in Table 6, Table 7, Table 8, Table 9, and Table 10. Across different models, our method consistently achieves near-perfect forget set error (approaching 100%), while maintaining minimal impact on retain and test set performance—typically increasing error by only 1%. These results demonstrate that AdaProb is a highly effective approach for poisoned data removal tasks.

Table 5: Selective unlearning results with ALL-CNN for the poisoned data removal task. Our method achieves higher forget rates while preserving overall model performance. "AdaProb w/ uniform" indicates that pseudo-probabilities are set to a uniform distribution, while "AdaProb w/ random" refers to pseudo-probabilities following a random distribution.

| Model | CIFAR-10 | | | Lacuna-10 | | |
| --- | --- | --- | --- | --- | --- | --- |
| | Test error | Retain error ($\downarrow$) | Forget error ($\uparrow$) | Test error | Retain error ($\downarrow$) | Forget error ($\uparrow$) |
| Retrain | 16.71 | **0.00** | 25.67 | 1.60 | **0.00** | 0.67 |
| Original | **16.43** | 0.00 | 0.00 | 1.53 | 0.00 | 0.00 |
| Finetune | 16.50 | 0.00 | 0.00 | 1.43 | 0.00 | 0.00 |
| Fisher | 21.39 | 4.00 | 13.00 | 1.87 | 0.01 | 0.00 |
| NegGrad+ | 21.36 | 3.23 | 45.33 | 2.77 | 0.40 | 8.67 |
| CF-k | 16.29 | **0.00** | 0.00 | 1.53 | **0.00** | 0.00 |
| EU-k | 17.62 | 0.11 | 0.33 | 1.83 | 0.00 | 0.00 |
| Bad-T | 22.43 | 10.13 | 1.67 | 4.90 | 1.34 | 0.67 |
| SCRUB | 16.55 | **0.00** | 20.33 | 2.07 | **0.00** | 1.67 |
| **AdaProb w/ random** | 17.00 | **0.00** | 86.00 | **2.20** | **0.00** | 64.00 |
| **AdaProb w/ uniform** | 16.60 | **0.00** | **95.00** | 2.80 | 0.42 | **68.00** |

Table 6: Unlearning results on ALL-CNN for the poisoned data removal task. Our method gets the highest forget error with little influence on model performance (retain error). "AdaProb w/ uniform" indicates that pseudo-probabilities are set to a uniform distribution.

| Model | CIFAR-5 | | Lacuna-5 | |
| --- | --- | --- | --- | --- |
| | Retain error ($\downarrow$) | Forget error ($\uparrow$) | Retain error ($\downarrow$) | Forget error ($\uparrow$) |
| Retrain | 0.13 | 28.80 | **0.00** | 4.67 |
| Original | 0.17 | 0.00 | **0.00** | 0.00 |
| Finetune | 0.04 | 0.00 | 6.63 | 19.33 |
| Fisher | 31.83 | 15.20 | 51.09 | 39.33 |
| NTK | 0.17 | 13.6 | **0.00** | 3.33 |
| NegGrad+ | 0.56 | 36.00 | 0.14 | 12.00 |
| CF-k | **0.00** | 0.00 | **0.00** | 0.00 |
| EU-k | 3.23 | 8.00 | **0.00** | 0.00 |
| Bad-T | 9.68 | 10.67 | 2.32 | 0.00 |
| SCRUB | 0.08 | 40.80 | **0.00** | 25.33 |
| **AdaProb w/ uniform** | 1.05 | **68.00** | 1.47 | **78.00** |

# B PRIVACY PROTECTION

We conducted additional experiments on privacy protection tasks shown in Table 11, evaluating forget, retain, and test set errors. Our results show that AdaProb achieves performance nearly identical to the retrained model across all sets, providing strong evidence of effective privacy protection.

# C KL DIVERGENCE WITH TEST SET INPUTS

In addition to calculating KL divergence on the forget set, we investigated the model's generalization ability through additional experiments on the test set. The results in Table 12 show that AdaProb achieves lower KL divergence compared to SCRUB when measured against the retrained model, indicating that AdaProb produces an unlearned model that more closely resembles the ideal retraining baseline. Also, Figure 4 use t-SNE map to helps visualize the output distribution of retrain, SCRUB, and AdaProb.

# D PERFORMANCE ON LARGE MODELS

We conducted additional experiments on poisoned data removal using CIFAR-100. As shown in Table 13, AdaProb achieves significantly higher forget error compared to SCRUB, demonstrating superior unlearning performance.

Table 7: Class unlearning results with ResNet for the poisoned data removal task. Our method gets top performance in forgetting with little influence on model performance (retain error). "AdaProb w/ uniform" indicates that pseudo-probabilities are set to a uniform distribution.

| Model | CIFAR-10 | | Lacuna-10 | |
| --- | --- | --- | --- | --- |
| | Retain error ($\downarrow$) | Forget error ($\uparrow$) | Retain error ($\downarrow$) | Forget error ($\uparrow$) |
| Retrain | **0.00** | **100.00** | **0.00** | 99.75 |
| Original | **0.00** | 0.00 | **0.00** | 0.00 |
| Finetune | **0.00** | 0.00 | **0.00** | 0.00 |
| Fisher | 2.45 | **100.00** | **0.00** | **100.00** |
| NegGrad+ | 1.74 | 91.26 | **0.00** | 14.90 |
| CF-k | **0.00** | 0.03 | **0.00** | 0.00 |
| EU-k | **0.00** | 98.79 | 0.01 | 4.06 |
| Bad-T | 11.34 | 94.67 | 1.06 | 67.60 |
| SCRUB | 0.51 | **100.00** | 0.28 | **100.00** |
| **AdaProb w/ uniform** | 2.48 | **100.00** | **0.00** | **100.00** |

Table 8: Class unlearning results with ALL-CNN for the poisoned data removal task. Our method gets top performance in forget with little influence on model performance (retain error). "AdaProb w/ uniform" indicates that pseudo-probabilities are set to a uniform distribution.

| Model | CIFAR-10 | | Lacuna-10 | |
| --- | --- | --- | --- | --- |
| | Retain error ($\downarrow$) | Forget error ($\uparrow$) | Retain error ($\downarrow$) | Forget error ($\uparrow$) |
| Retrain | **0.00** | **100.00** | **0.00** | **100.00** |
| Original | **0.00** | 0.00 | **0.00** | 0.00 |
| Finetune | **0.00** | 0.00 | **0.00** | 0.00 |
| Fisher | 3.66 | 99.00 | **0.00** | 89.00 |
| NegGrad+ | 0.58 | 87.22 | **0.00** | 6.56 |
| CF-k | **0.00** | 0.00 | **0.00** | 0.00 |
| EU-k | 0.13 | **100.00** | 0.00 | 77.19 |
| Bad-T | 5.84 | 81.93 | 0.37 | 38.65 |
| SCRUB | 0.12 | **100.00** | **0.00** | **100.00** |
| **AdaProb w/ uniform** | 0.20 | **100.00** | **0.00** | **100.00** |

# E    EXPERIMENT DETAILS

This section presents the hyperparameters used in our experiments. Table14 details the pretraining configuration, while Table15 specifies the training parameters.

Table 9: Selective unlearning results with ResNet for the poisoned data removal task. Our method gets top performance in forgetting with little influence on model performance (retain error). "AdaProb w/ uniform" indicates that pseudo-probabilities are set to a uniform distribution.

| Model | CIFAR-10 | | Lacuna-10 | |
|---|---|---|---|---|
| | Retain error ($\downarrow$) | Forget error ($\uparrow$) | Retain error ($\downarrow$) | Forget error ($\uparrow$) |
| Retrain | **0.00** | 29.67 | **0.00** | 1.0 |
| Original | **0.00** | 0.00 | **0.00** | 0.00 |
| Finetune | **0.00** | 0.00 | **0.00** | 0.00 |
| Fisher | 2.88 | 3.00 | **0.00** | 0.00 |
| NegGrad+ | 4.10 | 53.70 | 0.90 | 13.00 |
| CF-k | **0.00** | 0.00 | **0.00** | 0.00 |
| EU-k | 0.40 | 23.70 | 0.00 | 0.00 |
| Bad-T | 14.53 | 34.67 | 3.26 | 0.33 |
| SCRUB | **0.00** | 70.33 | **0.00** | 4.67 |
| **AdaProb** | 0.01 | **100.00** | 5.39 | **100.00** |

Table 10: Selective unlearning results with ALL-CNN for the poisoned data removal task. Our method gets top performance in forget with little influence on model performance (retain error). "AdaProb w/ uniform" indicates that pseudo-probabilities are set to a uniform distribution.

| Model | CIFAR-10 | | Lacuna-10 | |
|---|---|---|---|---|
| | Retain error ($\downarrow$) | Forget error ($\uparrow$) | Retain error ($\downarrow$) | Forget error ($\uparrow$) |
| Retrain | **0.00** | 25.67 | **0.00** | 0.67 |
| Original | **0.00** | 0.00 | **0.00** | 0.00 |
| Finetune | **0.00** | 0.00 | **0.00** | 0.00 |
| Fisher | 4.00 | 13.00 | 0.01 | 0.00 |
| NegGrad+ | 3.23 | 45.33 | 0.40 | 8.67 |
| CF-k | **0.00** | 0.00 | **0.00** | 0.00 |
| EU-k | 0.11 | 0.33 | 0.00 | 0.00 |
| Bad-T | 10.13 | 1.67 | 1.34 | 0.67 |
| SCRUB | **0.00** | 29.33 | **0.00** | 1.67 |
| **AdaProb** | **0.00** | **100.00** | 0.00 | **88.12** |

Table 11: Unlearning results with ALL-CNN for the privacy protection task.

| Model | CIFAR-10 | | | Lacuna-10 | | |
|---|---|---|---|---|---|---|
| | Test error | Retain error | Forget error | Test error | Retain error | Forget error |
| Retrain | 16.71 | 0.00 | 26.67 | 1.50 | 0.00 | 0.33 |
| Original | 16.71 | 0.00 | 0.00 | 1.57 | 0.00 | 0.00 |
| Finetune | 16.86 | 0.00 | 0.00 | 1.40 | 0.00 | 0.00 |
| NegGrad+ | 21.65 | 4.54 | 47.00 | 3.60 | 0.87 | 14.33 |
| CF-k | 16.82 | 0.00 | 0.00 | 1.57 | 0.00 | 0.00 |
| EU-k | 18.44 | 0.32 | 0.33 | 3.90 | 0.76 | 0.00 |
| Bad-T | 22.43 | 10.13 | 1.67 | 4.90 | 0.67 | 1.34 |
| SCRUB | 17.01 | 0.00 | 33.00 | 1.67 | 0.00 | 0.00 |
| SCRUB+R | 16.88 | 0.00 | 26.33 | 1.67 | 0.00 | 0.00 |
| **AdaProb** | 18.05 | 0.00 | 25.35 | 1.05 | 0.00 | 0.05 |

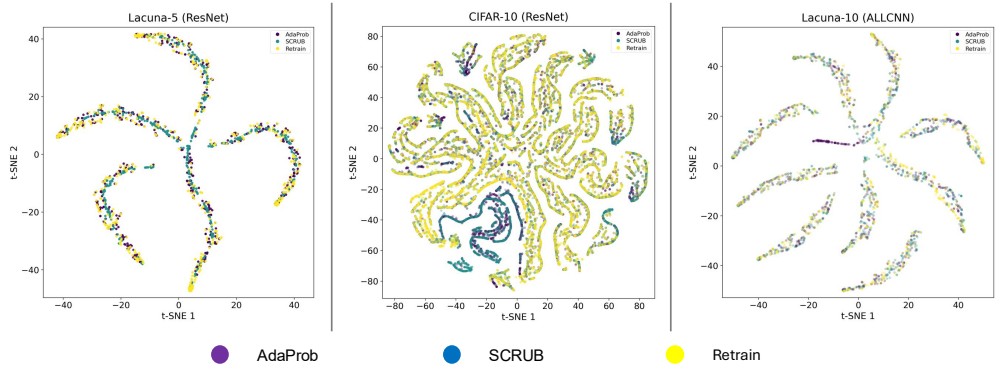

Figure 4: t-SNE map of output distributions on the test set for Retrain, SCRUB, and our unlearned model. Points closer to the Retrain cluster indicate stronger privacy (better unlearning).

Table 12: KL divergence between output distributions of unlearned and retrained models on test set inputs. A lower KL divergence indicates closer alignment with the retrained model's output distribution.

| Task | KL(AdaProb‖Retrain) (↓) | KL(SCRUB‖Retrain) (↓) |
|---|---|---|
| ResNet on Lacuna-5 | 0.044 | 0.21 |
| ResNet on Lacuna-10 | 0.76 | 0.88 |
| ResNet on CIFAR-5 | 0.056 | 0.11 |
| ResNet on CIFAR-10 | 0.18 | 0.20 |
| ALLCNN on Lacuna-5 | 0.10 | 0.09 |
| ALLCNN on Lacuna-10 | 0.22 | 0.23 |

Table 13: Performance on CIFAR-100 dataset with ResNet network

| Model | Forget error (↑) | retrain error (↓) |
|---|---|---|
| **AdaProb** | 98.25 | 0.00 |
| SCRUB | 5.19 | 0.00 |

Table 14: Hyperparameter for pretrained models

| Model | filter | learning rate |
|---|---|---|
| ALLCNN | 0.4 | 0.1 |
| ResNet | 1.0 | 0.1 |

Table 15: Hyperparameter for training models

| Model | filter | learning rate | weight decay | batch size | epochs | seed |
|---|---|---|---|---|---|---|
| ResNet (CIFAR5) | 0.4 | 0.001 | 0.1 | 128 | 31 | 3 |
| ALLCNN (CIFAR5) | 1.0 | 0.001 | 0.1 | 128 | 31 | 3 |
| ResNet (CIFAR5) | 0.4 | 0.001 | 0.1 | 128 | 31 | 3 |
| ALLCNN (Lacuna5) | 1.0 | 0.001 | 0.1 | 128 | 31 | 3 |
| ResNet (CIFAR10) | 1.0 | 0.01 | 5e-4 | 128 | 26 | 1 |
| ALLCNN (CIFAR10) | 1.0 | 0.01 | 5e-4 | 128 | 26 | 1 |
| ResNet (Lacuna10) | 1.0 | 0.01 | 5e-4 | 128 | 26 | 1 |
| ALLCNN (Lacuna10) | 1.0 | 0.01 | 5e-4 | 128 | 26 | 1 |

