# OpenReview forum: "AdaProb: Towards Efficient Machine Unlearning via Adaptive Probability"
_ICLR.cc/2026/Conference — ICLR 2026 Conference Withdrawn Submission_

### Official Review · Reviewer_5Qxt · 2025-10-31

**Soundness:** 2
**Presentation:** 3
**Contribution:** 2
**Rating:** 2
**Confidence:** 4

**Summary:**

This paper introduces a new approximate unlearning method called AdaProp. They perform unlearning via adaptive probability manipulation by directly adjusting the output-layer probabilities of a trained model. It replaces the forget-set predictions with pseudo-probabilities (e.g., uniform distributions or random probabilities) and fine-tunes the network to realize these distributions via backpropagation. The authors formulate the probability adjustment as a constrained optimization problem and solve its Lagrangian form via Coordinate Descent. The model’s weights are then updated via gradient descent by minimizing the KL between the optimal distributions and the model’s output. The paper takes experimental settings similar to SCRUB and shows it outperforms it with experiments on CIFAR-10/100 and the Lacuna dataset, and resnet and All CNN models.

**Strengths:**

**S1**. The paper separates poison unlearning from privacy cases which is a good practice because these two applications require different treatments.

**S2**. The paper is mostly written and presented clearly.

**Weaknesses:**

**W1**. Some more recent works, like [2], report results that outperform SCRUB in similar settings. It’s important to compare against the most recent methods to show SOTA results.

**W2**. The results are reported without any statistical test. The experiments should be repeated a few times, and the variance, at the very least, must be reported to get a sense of the significance. Without it, I personally can’t trust any results.

**W3**. The Related Work section is somewhat limited. It misses many recent works. Very weirdly, the authors report Bad-T in their results, but never introduce it as a baseline nor cite it in the literature!!

**W4**. Theorem 1 only has a proof sketch and the full proof is not provided.


**W5**. In table 2, the KL numbers are reported without a reference point. And mostly, the numbers are very close to each other across different models/datasets for AdaProb and SCRUB. How can one make sense of these numbers?

**W6**. The selected MIA for privacy evaluation is a very weak attack. In the SCRUB paper, they introduce a stronger and more reliable attack based on LiRA [2]. However, it is not evaluated in your experiments.

**W7**. In Figure 3, SCRUB seems even more expensive than full retraining. This is suspicious! In general, approximate unlearning methods should have a time budget of a fraction of Retrain. Even if they cannot achieve high performance, they should not be allowed to run too long.

**W8**. The introduced method includes two important optimizations, for probability adjustment and for model updates. There is no ablation of the effect of each optimization alone, nor are there results about their wall-clock/complexity time.

[1] https://openreview.net/pdf?id=3vXpZpOn29

[2] https://arxiv.org/abs/2112.03570

**Questions:**

**Q1**. Can you clarify the "over 20% improvement in forgetting error" in the abstract? The results in Table 1 suggest a much larger relative improvement!

**Q2**. The optimization in section 4.1 involves a matrix of size $N \times K$. How does this method's computational cost (specifically the optimization step, not the backpropagation) scale as the number of data points ($N$) and classes ($K$) becomes very large?

---

### Official Review · Reviewer_6sRD · 2025-10-31

**Soundness:** 3
**Presentation:** 3
**Contribution:** 2
**Rating:** 2
**Confidence:** 4

**Summary:**

This paper proposes a novel method named AdaProb, which allows unlearning effectively in privacy-preserving aspect. AdaProb first replaces the output probabilities of the model's final-layer with the pseudo-probabilities (uniform distribution) and this output is optimized to enhance the privacy. Then the model's weights are updated through backpropagation from the optimized probabilities. Empirical evaluation was also conducted in diverse settings to show competitive results of the proposed method against Membership Inference Attacks (MIA).

**Strengths:**

1. The paper is well-written with a clear scope for privacy-preserving.
2. New approach that optimizes both output distribution and model's parameters to unlearn effectively.
3. Diverse experiments were conducted to clarify the effectiveness in unlearning with privacy protection and computational efficiency.

**Weaknesses:**

1. The contribution lacks novelty, there is no theoretical support for the performance of unlearning in this work. Furthermore, the approach to optimize the model toward a given objective is found in various existing works, such as SCRUB (a baseline in this paper). The only innovative contribution of this paper is optimizing the output probabilities. Sections 4.2 and 4.3 are merely copied from preliminary knowledge.

2. The contribution is limited to a privacy-preserving manner and cannot be applied to other applications, such as removing poisoned knowledge in the model. In contrast, the objective of strategies like SCRUB is to remove knowledge from the forget set and retain knowledge from the retain set, which is suitable for the general purpose of unlearning. Therefore, other unlearning methods can be extended to cover all applications.

An example of an application that AdaProb cannot accommodate is preventing Backdoor Attacks (BA), as the objective of AdaProb is to align pseudo probabilities with the output of the model from the forget set (which is the poisoned knowledge in BA).

3. In all cases of purpose for unlearning, the result from the unlearned model should be close to the re-trained model. I saw in Table 1 that when you reported the result for the Poisoning Data Removal task, your objective is to maximize the forget error, which is misaligned with the unlearning objective. Your results for this criterion are also significantly different from the re-trained model's results.

If you only intend to protect your model from MIA, you should not classify your method as a machine unlearning technique.

4. The paper includes an experiment section for Poisoned Data Removal, but the setting is not clearly described. My understanding is that the authors used a normal setting (not a poisoning attack) for this section. If the authors are testing with a poisoning attack, there are no metrics to measure the actual remaining poisoning knowledge.

**Questions:**

I recommend the authors to read the weaknesses carefully and clarify:
1. The novelty of the proposed method, especially in section 4.2 and 4.3.
2. Applications and motivation of AdaProb when it is only applicable to privacy-preserving. Can you provide experiments for Backdoor Attack scenario?
3. Is your method an unlearning method?
4. Setting of section 5.4. Poisoned Data Removal.

---

### Official Review · Reviewer_D5dq · 2025-11-01

**Soundness:** 3
**Presentation:** 3
**Contribution:** 2
**Rating:** 4
**Confidence:** 4

**Summary:**

This paper introduces AdaProb (Adaptive Probability Approximate Unlearning), a novel machine unlearning method that operates in the output probability space of a trained model to efficiently forget specific data. AdaProb replaces the neural network’s final-layer output probabilities with pseudoprobabilities for data to be forgotten. These pseudo-probabilities follow a uniform distribution to maximize unlearning. These optimized probabilities are then backpropagated to update the model, effectively removing the influence of the forgotten samples while preserving utility. Experiments on multiple image classification benchmarks demonstrate that AdaProb achieves over 20% higher forgetting effectiveness in data-poisoning scenarios, maintains near-random accuracy under membership inference attacks, and reduces computational cost by more than half compared to the state-of-the-art SCRUB method.

**Strengths:**

1. Instead of simply overwriting forget-sample predictions with a uniform vector, the method solves a KL-regularized problem that makes the refined probabilities of the forget set similar to the model’s overall output distribution.
2. The paper correctly distinguishes two MU scenarios. For poisoning, it is fine to push forget samples to uniform and backpropagate directly. For privacy, the paper switches to the refined optimization that keeps retain predictions close to the original model.
3. The method consistently outperforms baselines in forgetting efficacy and efficiency, with comprehensive ablations on λ, architecture, and unlearning scale.

**Weaknesses:**

1. While AdaProb introduces KL-based refinement scheme, its overall two-stage pipeline closely mirrors that of SCRUB, resulting in limited conceptual novelty.
2. Theorem 1 only asserts convergence of a convex problem without rate or stability analysis. It also lacks evidence that the weight update preserves unlearning guarantees or that violating column constraints would increase MIA vulnerability.
3. There is no comparison to recent MU baselines, weakening the claim of superior or state-of-the-art performance.
4. The paper validates privacy only through a loss-based membership inference attack (MIA), without considering stronger or more adaptive attacks such as the relearn attack.

**Questions:**

see the comments above

---

### Official Review · Reviewer_W9nM · 2025-11-02

**Soundness:** 1
**Presentation:** 1
**Contribution:** 1
**Rating:** 2
**Confidence:** 5

**Summary:**

This paper proposes AdaProb, a novel method designed to address two primary limitations of existing machine unlearning approaches: the persistence of residual information (incomplete removal) and high computational overhead. The core mechanism involves inducing forgetting by replacing the final-layer output probabilities for the forget set with pseudo-probabilities (e.g., a uniform distribution). In parallel, the method aims to preserve model utility by constraining the model to maintain its original output distribution for the retain set. The model's weights are then updated based on these new target distributions. The authors claim this approach maximizes data removal for the forget set while simultaneously enhancing privacy (e.g., resistance to Membership Inference Attacks). Empirically, the paper reports that AdaProb achieves over a 20% improvement in forget error and reduces computational costs by more than 50% compared to state-of-the-art approximate unlearning methods.

**Strengths:**

-	The paper's core methodological contribution is its constrained optimization framework (Sec 4.1). This approach successfully achieves the dual objectives of removing information from the forget set while maintaining performance on the retain set, and, most importantly, is shown to be highly effective in defending against Membership Inference Attacks (MIA).

-	The method demonstrates overwhelming computational efficiency. When compared to the state-of-the-art baseline (SCRUB) and the gold standard (Retrain), AdaProb completes the unlearning task in less than half the computation time (Sec 5.6), proving its practical viability.

-	The KL Divergence analysis (Sec 5.5, Table 2) and corresponding visualizations (Fig. 2) provide strong evidence that the output distribution of the AdaProb-unlearned model is the most similar to the 'Retrain' model's distribution, surpassing the SOTA SCRUB baseline in approximating the true gold standard.

**Weaknesses:**

-	Limited Architectural Scope: The experiments are conducted exclusively on CNN-based models (ResNet-18, ALL-CNN). This makes it impossible to verify the method's scalability or applicability to other dominant architectures, such as Transformers.

-	Significant Lack of Methodological Novelty: The proposed method does not introduce a fundamentally new unlearning mechanism but rather presents a variation of existing finetuning approaches. This is evident in both of the paper's main scenarios:The method used for Poisoned Data Removal (Sec 5.4), which sets the forget set's target to a 'Uniform/Random' distribution and then finetunes, is conceptually identical to the existing 'Random Labeling' baseline. This cannot be considered a methodological contribution.The 'constrained optimization' framework (Sec 4.1), claimed as the core contribution for Privacy Protection, also relies on standard finetuning (gradient descent) as its actual removal mechanism (Sec 4.3, Eq 8 & 9). The complex optimization process (Sec 4.1) is merely a "pre-calculation" step to "generate a target"  for this finetuning.

-	Limited Applicability to Complex Datasets: The experiments are confined to single-label datasets (CIFAR, Lacuna). The method's effectiveness on more complex datasets, such as ImageNet or multi-label datasets (e.g., COCO)—where multiple classes are mixed within a single image—is questionable. It is highly likely that attempting to forget one class in such a setting would either fail to unlearn completely or negatively impact the retained set.

-	Overly Restrictive Baseline Comparisons: The paper's claim that SCRUB is the SOTA is used to justify limiting the comparative graphs and t-SNE visualizations to only Retrain, SCRUB, and AdaProb. This is excessively restrictive. Richer insights and interpretations would have been possible if other baseline methods (e.g., Fisher, NTK, NegGrad+) were also included in these visual comparisons.

-	Missing Qualitative Visualizations: Given that the Lacuna-5 dataset consists of face images, a crucial piece of evidence is missing: visualizations, such as Grad-CAM heatmaps, comparing the model's attention before and after unlearning. This would be essential to qualitatively demonstrate that the model has indeed "forgotten" the target class.

-	Minor Presentation Issues: While not a critical flaw, the main illustrative image (Figure 1) appears to be slightly cut off on the left side, and the spacing of numbers and shapes is inconsistent. This should be corrected for professional presentation and clearer communication to the reader.

**Questions:**

Clarity is required for Section 5.4, "POISONED DATA REMOVAL." Referring to what appears to be clean forget data as "Poisoned data" is misleading and can cause significant confusion for readers.

This terminology is easily confounded with actual adversarial poisoning scenarios, such as label-flipping or backdoor attacks. If the authors did not experiment on such scenarios, it would be more accurate and appropriate to define this set simply as the "forget set" or a "subset of the whole dataset" to avoid misrepresentation.

---

### Author Response · Authors · 2025-12-01

We thank the reviewer for their time and valuable feedback. After discussion, we have decided to withdraw our submission. We agree with the reviewer’s suggestion that a new baseline is required for a fair comparison and that a stronger membership inference attack evaluation is necessary. We will incorporate this feedback to revise our work.

---

### Note · Authors · 2025-12-01

I have read and agree with the venue's withdrawal policy on behalf of myself and my co-authors.